# Public Health Insurance, Non-Farm Labor Supply, and Farmers’ Income: Evidence from New Rural Cooperative Medical Scheme

**DOI:** 10.3390/ijerph16234865

**Published:** 2019-12-03

**Authors:** Jin Liu, Yufeng Lu, Qing Xu, Qing Yang

**Affiliations:** 1Research Institute for Agriculture, Farmer and Rural Society in China, Shanghai University of Finance and Economics, NO.777, Guoding Road, Yangpu District, Shanghai 200433, China; liu.jin@mail.shufe.edu.cn; 2Institute of Finance and Economics, Shanghai University of Finance and Economics, NO.777, Guoding Road, Yangpu District, Shanghai 200433, China; luyufeng421@163.com (Y.L.); yangqing@163.shufe.edu.cn (Q.Y.)

**Keywords:** New Rural Cooperative Medical Scheme (NRCMS), non-farm labor supply, farmers’ income, non-portability

## Abstract

The major source of income of Chinese farmers is non-farm income, especially wages and salaries. Based on the economics theory of health and healthcare, their non-farm labor supply behavior could be affected by health insurance policies. The work presented in this paper focuses on the impact of the New Rural Cooperative Medical Scheme (NRCMS) on farmers’ non-farm labor supply behavior in China. A four-part model regression approach was used to examine the relationship. Our dataset comprised of 8273 people, aged 45 or above, from the China Health and Retirement Longitudinal Study (CHARLS) conducted in 2011 and 2013. The empirical results showed that NRCMS significantly reduced non-farm labor force participation and employment. Compared to non-participants of the NRCMS, the non-farmer labor time of these participants reduced, but the supplementary medical insurance and immediate reimbursement of the NRCMS increased the participants’ non-farm labor time. Our results have contributed to the reform of China’s public health insurance and farms’ income growth, and it would be necessary to actively promote immediate reimbursement, gradually simplify reimbursement procedures for medical treatment in non-registered places, and eliminate the non-portability of NRCMS.

## 1. Introduction

Since 2000, China’s agriculture and rural economy have developed rapidly. China, for sure, will face a slowdown in the economic growth as the sustained and stable promotion of farmers’ income is facing a new development environment, which needs to adapt to new changes and challenges. On the one hand, the numbers of farmers making a living on agriculture is becoming less and less. By the end of 2013, pure farmers accounted for 39.65% of all farmers, while non-farmers and part-time farmers accounted for more than 60% [1]; on the other hand, the business income of per capita disposable income of rural households has declined year by year, especially the share of agricultural income has declined from more than 50% to about 26% and that of wages has increased from 22% in 1995 to 43.5% in 2012. Therefore, increasing non-farm labor supply and non-farm income has become the key component to increasing farmers’ income. 

Theoretically, as an important part of the social security system, the public health insurance scheme helps to protect the physical and mental health of laborers, improve labor productivity, and then promote development and increase residents’ income [2,3,4,5]. Although a large number of previous empirical studies have discussed the relationship between public health insurance and the labor market [3,4,5,6,7,8,9,10,11,12,13], due to different understanding of the actual situation, different research data, and different research methods, the conclusions are quite different and unsystematic. At present, as farmers’ non-farm income gradually increases in China, the New Rural Cooperative Medical Scheme (NRCMS) that aims to improve rural residents’ health and welfare is a good material for in-depth study of such issues.

Previous literature on farmers’ income showed that the new agricultural management mode and scale operation helped to increase farmers’ income [14,15]. With the industrialization and urbanization in China, the role of land in farmers’ income is being blurred by degrees [16], and the proportion of wages and salaries of farmers’ income is becoming higher and higher. Therefore, a lot of researchers mainly focus on human capital, which is closely related to wages and salaries. Undoubtedly, as a crucial part of human capital, health is not only a significant human “capability” [17], but also has an instrumental value in improving labor productivity, increasing personal income, and expanding economic participation [18]. Many studies have pointed out that the improved health status not only increases farmers’ labor supply [8,13,19,20,21,22,23,24], but also actively reduces rural poverty or increases farmers’ income [25,26,27].

On the other hand, many scholars have evaluated the effect of NRCMS on medical expenditure, living consumption, income, health status, and labor supply. Some believed that NRCMS did not significantly reduce the actual medical expenditure and the incidence of serious illness expenditure of farmers [28,29,30]. Others insisted that NRCMS promoted farmers’ household consumption, which increased with the rising level of medical insurance [31]. In terms of the effect of NRCMS, Tan and Zhong [32] used the household survey data of Jiangsu and Anhui provinces to find that the compensation of NRCMS was more designed for the sick population, and the compensation of the low-income group was higher than that of the high-income group. Similarly, Qi [33] pointed out that NRCMS could significantly promote the income of low- and middle-income farmers, but it needed a positive external economic environment as a supportive condition. Zhao et al. [34] investigated the impact of health insurance on the risk of obesity in rural China using longitudinal data from the China Health and Nutrition Survey (CHNS). Their results revealed that NRCMS participation had a significant positive impact on people’s tendency toward unhealthy lifestyles, such as high-fat food, cigarette smoking, and heavy drinking. More studies have verified that NRCMS played a significantly role in improving farmers’ health status [13,34], especially short-term health human capital [35]. Meng et al. [36] used data from the 2015 China Migrants Dynamic Survey (CMDS) to analyze the causal relationship between health insurance and the health of the senior floating population. The results implied that participation in the health insurance system significantly improved floating seniors’ self-rated health, and joining a place of settlement could improve the health of the floating senior population. Due to the non-portability and discriminatory characteristics of the current NRCMS policy, a large number of studies have found that it reduced farmers’ free mobility of participating in the labor market, thus “locking” the labor force in the place of household registration [37,38,39,40] or in agricultural production activities [41,42], thus forming a “lock-in effect” [43,44,45]. In fact, Fang et al. [46] argued that migration to urban areas limited the effectiveness of rural health insurance on hypertension management due to its non-portable nature.

It can be found from the literature that, although scholars have discussed methods to promote farmers’ income and focused on the actual effects of NRCMS policy, the following issues need to be further analyzed: First, in the investigation of how to promote farmers’ income, sufficient attention has not been paid to how NRCMS affects non-farm labor supply. Secondly, the employed data shows that previous research mainly focused on the survey between 2003 and 2009, but there is still a blank for relevant research after the reform of new health and healthcare in 2009. In fact, with the development of NRCMS, the government’s increase in financial investments, the improved relevant compensation standards, and the rural residents’ deeper understanding of NRCMS, further discussion is needed on whether and how the labor supply behavior has changed. Thirdly, existing research only distinguished whether participating in NRCMS has an impact on rural labor migration and regarded the participants as facing a homogeneous institutional structure, without examining the impact of heterogeneity of NRCMS on the labor market in terms of payment structure, such as starting line, reimbursement ratio, and capping line. Therefore, the existing research cannot fully and objectively outline the effect of NRCMS policy on farmers’ labor supply. Will farmers’ labor supply be affected by NRCMS? If the answer is yes, then what is the mechanism? Obviously, this field still needs further analysis through theoretical and empirical research.

As mentioned before, the current NRCMS policy requires farmers to participate in insurance and medical treatment in the place of household registration, which is strongly non-portable and handicaps insured persons’ migration [38,40], so that farmers are “locked” in agricultural activities, forming the “lock-in” effect [40,41,42]. Moreover, under the current three-level reimbursement policy of NRCMS, the participants enjoy higher reimbursement level in the place of household registration, which will increase the opportunity cost of farmers going out to work, keep them working in the county, and increase their willingness to return, i.e., there is a “pull back” effect. Empirical studies also suggested that public health insurance policies can reduce the labor participation rate of workers [8,11,43,44,45,46], which does not help to increase farmers’ income. By investigating the influence of NRCMS on farmers’ decision-making in agricultural production, Zhang et al. [42] found that under the realistic trend of agricultural concurrent and sideline, farmers’ hope to keep the original amount of land and stay there does not increase non-farm labor supply, thus affecting their income, but the effect was not significant.

The primary objective of this study is to investigate whether the influence of NRCMS on farmers’ labor supply is reflected in the level of non-farm labor supply? If so, does it make a difference and how? If it can significantly increase farmers’ non-farm labor supply, this will undoubtedly provide a new way to increase their income. 

This study contributes to the literature in two ways. First, by using the four-part model, this paper makes an empirical analysis of the influence of NRCMS on farmer’ non-farm labor supply and its mechanism by investigating the China Health and Retirement Longitudinal Study (CHARLS) conducted in 2011 and 2013. Second, the results could be helpful for further promoting the development and improvement of public health insurance and growth of farmers’ income in China.

## 2. Theory and Mechanism of NRCMS on Farmers’ Non-Farm Labor Supply

### 2.1. Theory

The aim of this paper was to examine the effect of NRCMS on non-farm labor, i.e., farmers’ willingness to participate in labor and whether their labor supply time was affected by NRCMS. Therefore, it was necessary to analyze farmers’ labor supply behavior through the theory of farmers’ labor behavior decision-making.

The farm-household model, first proposed by Becker [47], analyzes farmers’ labor behavior decision-making. The model assumes that farmers allocate time for leisure, agricultural labor, and non-farm labor according to the principle of maximizing family utility. This study introduced NRCMS and health loss into the budgetary constraints and production conditions of the farm-household model as follows:(1)MaxU=U(Y,E),
where *U, Y*, and *E* represent utility function, income, and leisure respectively. The function of time and wealth constraints that individuals face are as follows:(2)$$(1-\delta )T=LS^{f}+LS^{n}+E,[LS^{f},LS^{n},E>0,\delta \in (0,1)],$$
(3)$$Y=w_{n}LS^{n}+p_{Y}Y_{f}-p_{X}X_{f}+E(P)+V,$$
where T represents the whole family time and δ represents the health loss. The worse the health condition, the larger the *δ* value becomes and the less time an individual can spend on agricultural labor, non-farm labor, and leisure work. $LS^{f}$ represents agricultural working time and $LS^{n}$ represents non-farm labor time. $LS^{n}>0$ means farmers participate in non-farm labor and $LS^{n}=0$ means farmers do not participate in the non-farm labor market. E represents leisure; Y represents the total income; $w_{n}$ represents non-farm wage rate; $p_{Y}$ and $p_{X}$ represent the price of agricultural output and the cost of agricultural input, respectively; $Y_{f}$ and $X_{f}$ represent agricultural output and agricultural input, respectively; E(p) represents the expected income of NRCMS; and V represents other transfers of income.

The optimization problem was to solve the maximization problem in Equation (1) under constraints of Equations (2) and (3) and obtain the optimal time investment. This study no longer carried on the optimization solution of the theoretical model, but focused on the theoretical framework to explain the mechanism of NRCMS on non-farm labor decision-making. The theoretical analysis of the farmer household model cannot get the direction of the role of NRCMS on the non-farm labor supply. Therefore, the labor participation model and the labor supply model based on the above model should be built for an empirical analysis of the supply effect of NRCMS on non-farm labor, so as to get the specific direction of action and the influence.

### 2.2. Theoretical Framework

The main objective of NRCMS is to secure farmers’ access to basic medical services, alleviate their poverty caused by illness, and slow down their return to poverty due to illness. Therefore, the scheme can influence farmers’ non-farm labor supply behavior through improving health status and reducing the burden of family medical expenditure as follows:

Firstly, NRCMS increases the non-farm labor supply by improving the health status of farmers. Health is not only a significant part of human well-being, but also the basis of all economic activities. In the case of insufficient supply of informal social security system in rural areas, NRCMS ensures farmers’ access to basic medical services and helps to improve their health status. This improvement will not only reduce the time lost due to illness, but also increase labor productivity, labor supply, and leisure time, so as to increase their labor capacity in the long term. In other words, NRCMS will help to increase farmers’ non-farm labor supply and then increase the non-farm income and the total family income by improving their health status. Secondly, NRCMS reduces the burden of household medical expenditure (including direct medical expenses and indirect costs, such as loss of working time due to illness) and its uncertainty by implementing compensation policies of hospitalization and outpatient service, so that farmers’ income remains unchanged or relatively improved (It is worth noting that because the medical and health service market has the characteristics of information asymmetry, supplier-induced demand, externalities, and health is of great value, participating farmers may increase their medical expenditure in the short term (especially for unnecessary medical drugs and services), but with the expansion of the compensation scope of NRCMS, the increase in the catalog of essential drugs and the establishment and improvement of the preventive and rehabilitative medical service market, the medical expenditure of farmers and their families will decrease in the long run.). Therefore, on one hand, farmers can use the economic resources originally used to cope with health risks for human capital investment or production investment, and further increase the non-farm labor supply and labor productivity, thereby making it easier to obtain employment opportunities and increase income in the labor market (Of course, there is also the possibility that after the implementation of NRCMS, some farmers may shift their expenditure from disease risk prevention to unreasonable consumption, such as feasting, drinking, whoring, and gambling.). On the other hand, with the increase in household income, farmers can also increase expenditure on nutrition, preventive health, and skill training, so as to maintain good health and labor capacity, and the improvement in health and labor capacity will in turn promote income growth. In addition, with the improvement of family income, farmers can increase social life expenditure, strengthen their social network, and expand non-farm employment channels, so that they can get more non-farm employment opportunities and promote income growth.

In addition, when examining the influence of NRCMS on farmers’ non-farm labor supply behavior, the following phenomena should be attended to: on one hand, because the current NRCMS requires farmers to participate in insurance and medical treatment in the place of household registration, there is a big difference in medical treatment reimbursement between a registered place and a non-registered place, which will reduce farmers’ cross-region migration to some extent, especially cross-province migration, “locking” the labor force in the place of household registration and forming a "lock-in effect". On the other hand, NRCMS carries out compensation policy by levels; however, the corresponding medical system has not yet been established, the distribution of medical insurance resources is not balanced, and there are problems in the implementation of NRCMS, such as the division of mutual recognition of diagnosis and treatment results in designated medical institutions in different regions, the inconsistency of the catalog and scope of reimbursement for essential drugs, and the participation of farmers in non-registered places. Farmers in need of reimbursement in non-registered areas will face complex procedures, as well as economic losses and high transaction costs resulting from reduced working hours [35]. From this point of view, NRCMS is a non-portable and discriminatory public health insurance policy. For one thing, it will not only impose geographical restrictions on participating farmers’ medical treatment, but also hinder the free migration of participants in the labor market, so that the labor force is "locked" in the place of household registration. On the other hand, under the current NRCMS three-level reimbursement policy, the participating farmers enjoy higher reimbursement in the place of household registration than in other places, which will increase the opportunity cost for farmers to go out to work, make farmers prefer to work in the hometown, and increase the willingness of returning workers, i.e., there is a “pull-back” effect.

## 3. Methods, Variables, and Data 

### 3.1. Methods and Variables

#### 3.1.1. Methods

This paper divides the discussion of farmers’ non-farm labor supply behavior into the following parts: the first part is about their participation in non-farm labor decision-making, namely, “whether to participate in non-farm labor”; the second part is about their decision-making on non-farm labor supply (Non-farm self-employment also includes helping with one’s household management.), i.e., “whether to choose employment or non-farm self-employment”. The last part is the working time of the employed and the non-farm self-employed. Therefore, a four-part model [48,49] was used to analyze farmers’ non-farm labor supply behavior. The specific four models are as follows:

Model 1: Probability model of the choice behavior of non-farm labor participation. This is the probability model of non-farm labor supply in a given period of time, which distinguishes whether there is non-farm labor supply or not. In this model, the probability ($Labor_{1i}$) of non-farm labor supply (including employment and non-farm self-employment) occurring within a given period of time is expressed as follows:(4)$$Labor_{1i}=X_{i}\beta_{1}+\varepsilon_{1i},$$
where $Labor_{1i}>0$ means non-farm supply is positive; otherwise, it is 0. $\varepsilon_{1i}\sim N(0,1)$, this part takes all samples and those participating in NRCMS as the research object, investigating the influencing factor of farmers’ decision-making on non-farm labor participation. Because the decision-making of the non-farm labor participation is divided into "choosing non-farm employment" and "not choosing non-farm employment", it belongs to the variable of "(0–1)", so the binary Probit regression model is adopted.

Model 2: Probability model of non-farm employment type. According to the samples of non-farm employment, the model examines whether they choose to be employed or self-employed. This model also belongs to the "0–1" variable. The model is expressed as follows:
(5)$${\mathrm{Labor}}_{2i}=X_{i}\beta_{2}+\varepsilon_{2i},$$
where $\varepsilon_{2i}\sim N(0,1)$. Just like Model 1, the dependent variable is binary variable, so the binary Probit regression model is adopted again.

Model 3: Model of employed samples’ labor supply time. The model is expressed as follows:(6)$$Day_{i}{|}_{Labor_{1i}>0\&Labor_{2i}=1}=X_{i}\beta_{3}+\varepsilon_{3i},$$
where $\varepsilon_{3i}\sim N(0,\sigma_{\varepsilon_{3i}}^{2})$.

Model 4: Model of non-farm self-employed samples’ labor supply time. The model is expressed as follows:(7)$$Day_{i}{|}_{Labor_{1i}>0\&Labor_{2i}=0}=X_{i}\beta_{4}+\varepsilon_{4i},$$
where $\varepsilon_{4i}\sim N(0,\sigma_{\varepsilon_{4i}}^{2})$.

Because the dependent variables in Models 3 and 4 belong to continuous variables, the least square method is used for regression. In Equations (4)–(7), $Labor_{i}$ refers to individual i’s decision-making on non-farm labor participation, including whether to participate in non-farm employment and whether to choose employment in the participation; $Day_{i}$ refers to individual *i*’s labor supply time when there is non-farm employment; $X_{i}$ represents variables of social demographic and economic characteristics of NRCMS, individual characteristics, and family characteristics; and *ε* represents error term.

#### 3.1.2. Variables

##### Labor Supply 

We examined three labor supply indicators: (i) Non-farm labor participation; (ii) the types of non-farm employment in samples of non-farm labor, i.e., employment and non-farm self-employment (including helping with household management). In CHARLS, the labor supply was obtained by asking participants questions, such as “Do you have two (or more) non-farm jobs besides agricultural production and operation?” and “What are the main jobs (i.e., the longest working hours) in these jobs? Are they wage-earning jobs engaging in individual or private economic activities, or non-wage jobs to help with household management?” The paper analyzed farmers’ non-farm labor participation and employment type choice behavior; and (iii) labor time, i.e., the total working time in the current year. Data were obtained by asking “How many months have you worked in the past year?”, “How many days do you usually work per week?”, and “How many hours do you usually work per day?” It should be noted that in the CHARLS questionnaire, employment was divided into two types: one was from the employing organization and the other was from the dispatching organization, but the survey did not ask the respondents about the time supply of labor dispatch. Therefore, this paper only considered the labor supply behavior of samples with employing organization, when investigating the factors influencing labor time.

##### Institutional Variables of NRCMS

The key explanatory variable in this study was the institutional variable of NRCMS. In CHARLS, information about NRCMS was obtained mainly through the following questions:

(1) “Participating in NRCMS or not”. According to the data of the two CHARLS surveys, the proportion of farmers participating in NRCMS was about 90% in the survey year, and 10% of them did not. Therefore, by distinguishing whether or not farmers participated in NRCMS, this paper examined the differences in the non-farm labor supply between participants and non-participants. 

(2) “Participating in NRCMS supplementary medical insurance (such as major illness medical treatment) and reimbursement methods (i.e., when seeking medical treatment, medical expenses should be reimbursed immediately or first paid by oneself and then reimbursed) or not”. According to the current policy, NRCMS’ payment structure and level, the reimbursement of hospitals at all levels, starting line, top line, and drug reimbursement vary with different regions; at the same time, only within the reimbursement scope can the corresponding reimbursement ratio be enjoyed, and the annual cumulative reimbursement expenses cannot exceed the annual top line. It implies that simply considering whether farmers participate in NRCMS will weaken the quantitative evaluation of the effect of NRCMS as a public policy. Therefore, this study also investigated how NRCMS policies affected farmers’ non-farm labor supply behavior from the dimensions of NRCMS supplementary medical insurance (such as major illness medical treatment) and reimbursement methods combined with the content of CHARLS questionnaire.

##### Other Variables

This study controlled for health status, annual and provincial dummy variables, age, education level, gender, marital status, family size (Main respondents and their spouses are excluded.), annual net income per household, land ownership (mainly arable land or woodland), etc. Health status was measured by self-assessment of health status. Additionally, provincial dummy variables reflected the differences of NRCMS participation rate and related compensation policies among provinces (such as the structure of hospitalization compensation); on the other hand, they also reflected provincial location characteristics and cultural system background. 

### 3.2. Data 

The primary data used in this research work were drawn from the China Health and Retirement Longitudinal Study (CHARLS). CHARLS had received critical support from the Peking University, the National Natural Science Foundation of China, the Behavioral and Social Research Division of the National Institute on Aging, and the World Bank. CHARLS was a nationally representative longitudinal survey of persons, aged 45 or above, and spouses, including assessments of social, economic, and health circumstances of community residents in China [50]. All data will be made public one year after the end of data collection. CHARLS adopted multi-stage stratified PPS sampling. 

At present, on one hand, young and middle-aged people in China generally leave agriculture, and the proportion of farmers over 51 years old in agricultural labor force has exceeded 32% (Data sources: Office of the Leading Group of the Second National Agricultural Census of the State Council, National Statistical Bureau of the People’s Republic of China (Editor): Comprehensive Summary of Data of the Second National Agricultural Census of China, China Statistics Publishing House, 2008.). On the other hand, under the current realistic background that farmers leave the countryside and seek to go out to work, the proportion of young agricultural labor in China is declining year by year—not only the average age of migrant workers has increased from 35.5 years in 2010 to 38.3 years in 2014, but also the proportion of those over 50 years old has increased from 12.9% to 17.1% (Data sources: Bureau of Planning: National Survey Report on Migrant Workers in 2014.). It means rural middle-aged and elderly people have become an important part of China’s labor market. Moreover, for a long time, rural residents in China have not been restricted by the old-age security system and the compulsory retirement age of official departments in terms of labor supply, but have been working in good health until they are unable to.

The CHARLS questionnaire included the following modules: demographics, family structure/transfer, health status and functioning, biomarkers, healthcare and insurance, work, retirement and pension, income and consumption, assets (individual and household), and community level information. The baseline national wave of CHARLS was fielded in 2011 and included about 10,000 households and 17,500 individuals in 150 counties/districts and 450 villages/resident committees (or villages) from 28 provinces. Furthermore, the CHARLS respondents were followed up every two years, using a face-to-face computer-assisted personal interview. Since 2011, the national follow-up survey data of CHARLS for 2013, 2015, and 2017 were publicly released. More detail could be found on the website and downloaded at http://charls.pku.edu.cn/pages/Data/111/en.

Since we were interested in exploring the effect of NRCMS on farmers’ non-farm labor supply, we restricted our attention to the sub-sample of respondents, aged 45 or above. After eliminating the missing variables, the final sample contained 8273 individuals.

#### Descriptive Results

Table 1 gives descriptive statistics of the major independent variables. Results showed that more than 17% of the 8237 samples had non-farm employment in the year before the interview, and more than 90% of the respondents had participated in NRCMS, indicating that NRCMS basically achieved the goal of covering all farmers. In the meantime, 701 and 502 of the participating farmers chose to be employed and non-farm self-employed, with an average annual working time of about 1710 hours and 1925 hours, respectively.

From the participating sample, there was a very small proportion of participants in NRCMS supplementary insurance, which was less than 5%, indicating that the promotion and support of the supplementary insurance of NRCMS, especially the major illness insurance, should be strengthened in the future; As for reimbursement methods, about 70% of the participants, both employed and non-farm self-employed, had to pay by themselves before reimbursement.

In addition, Table 1 also reports the mean values of other relevant variables. For instance, in regard to age, the average age of the employed was about 54 years, while that of the non-farm self-employed was about 55 years. To some extent, this reflects the current trend of rural labor transfer in China, i.e., the average age of the farmers who live on agriculture is gradually rising.

As for the annual net income per capita, the income of each sample group was relatively low, ranging from 5200 to 6300 yuan, reflecting the urgency of increasing farmers’ income. In terms of educational level, more than 60% of the agricultural labor had received primary and higher education. Regarding the self-rated health status, more than 80% of the respondents, whether employed or non-farm self-employed, believed that they were healthy. Marital status, family size, land ownership, and other variables showed no obvious difference between different samples.

## 4. Empirical Results

Table 2 and Table 3 display the regression results of farmers’ non-farm labor supply behavior model.

(i) Regression analysis of non-farm labor participation probability model. The regression results suggested that the probability model of non-farm labor participation passed the joint test, and the regression was significant overall.

Firstly, the paper discusses the most concerned part, the NRCMS variables. In all samples, “whether to participate in NRCMS” reduced the probability of farmers participating in non-farm labor, and was statistically significant at the level of 0.1. The results of marginal effect calculation suggested that, when other variables remained unchanged, the willingness of participating farmers to participate in non-farm labor was reduced by 1.6% compared to farmers who did not participate in NRCMS. The reason is that the current NRCMS policy requires participating farmers to pay fees in the place of household registration and to seek medical treatment and reimbursement in local designated health and medical institutions, which actually imposes geographical restrictions on participation and reimbursement, thus affecting farmers’ decision-making on non-farm labor participation; meanwhile, although the pilot work of NRCMS on medical treatment and reimbursement in non-registered places has been carried out, there are still many difficulties. Farmers participating in medical treatment in non-registered places will face complex reimbursement procedures, as well as economic losses and high transaction costs resulting from reduced working hours. This was supported by the regression results of the participating sample model: whether to participate in NRCMS supplementary insurance and whether to reimburse immediately had statistical significance at the level of 0.1, which also proved that the profits of participating farmers from NRCMS affected their willingness to participate in non-farm labor.

With regard to individual and family characteristics, in the two models of all samples and the participating sample, the non-farm participation willingness of older people, women, and those with spouses was significantly lower due to their weaker labor ability or stronger family tie. As for the education level, compared with those who had only received education below primary school, farmers who had received education above primary school will significantly increase their non-farm labor participation rate, which could be attributed to their better education and relatively greater knowledge, skills, and openness to ideas. The results of marginal utility suggested that, with other variables unchanged, the non-farm labor participation rate of farmers with primary school education and above would increase by 4.1% and 3.7%, respectively. Annual net income per capita had a significant positive effect on farmers’ willingness to supply non-farm labor, which reflected that farmers paid more attention to non-farm income, broadened their non-farm employment channels, and diversified their income sources.

(ii) The regression analysis of the probability model of non-farm labor type choice. In Model 3, the variables of participation, age, sex, marital status, and land ownership were tested by statistical significance, and other variables were not the main factors affecting the non-farm labor type choice behavior of non-farm workers.

The paper focused on the analysis of the impact of NRCMS on farmers’ non-farm labor type choice behavior. Firstly, whether to participate in NRCMS significantly reduced the probability of farmers who participated in non-farm labor to be employed. One explanation is the implementation of NRCMS as a partial reimbursement system, which stipulates that the expenses of participating farmers in different levels of hospitals should be reimbursed by different levels. The reimbursement ratio of hospitals in the county was significantly higher than that of hospitals outside the county, while the starting line was just the opposite. More importantly, there was segregation and non-convergence between different regions in the implementation of NRCMS. There was a complex reimbursement procedure in the rural cooperative medical management office of the registered place for medical expenses occurring in non-registered places, and farmers will also face economic losses and high transaction costs resulting from reduced working hours. As mentioned above, the participation and compensation of NRCMS had geographical limitations with prominent “non-portability” characteristics, so that participating farmers were locked in the place of household registration.

In addition, there was no statistical significance in whether to participate in supplementary insurance and reimbursement methods when investigating the employment probability of the participating sample, which needs further study.

(iii) Analysis of regression results of labor time supply model of employee samples. The regression results of the model (Table 3) suggested that the working time of men and those in good health will increase significantly, while the working time of older employees, those with a big family, and land-owning employees will decrease significantly. In terms of education level, compared with those employed with education below primary school level, those who had received primary school education and above will have more working hours, which indicated that the better educated people with more knowledge and skills were more likely to obtain employment opportunities in the labor market, and were willing to increase their working hours, so as to get more economic income.

Table 3 reports that as one of the concerns of this paper. NRCMS variables (including participation, supplementary medical insurance, and reimbursement methods) had no statistical significance on the working hours of employed farmers.

(iv) Analysis of regression results of labor time supply model for non-farm self-employment samples. The regression results of the model suggested that NRCMS variables (including participation, supplementary medical insurance, and reimbursement methods) had no statistical significance on the labor time of non-farm self-employed farmers.

Age and land ownership affected the labor time of non-farm self-employed farmers, while other variables had no significant impact. The empirical results indicated that age had a significant negative effect on the working time of non-farm self-employed farmers at the statistical level of 0.01; compared with farmers without land, the working time of non-farm self-employed farmers with land was significantly reduced.

## 5. Discussion and Policy Implication

Some literature had pointed out the importance of labor supply or off-farm employment on agricultural production and household income in rural China [51,52,53], but those ignored the role of public health. NRCMS is a critical part of China’s public health insurance. There are differences in the implementation of NRCMS within and outside the county, which weakens the effect of poverty reduction and income increase and affects farmers’ benefits from the scheme, thus influencing farmers’ labor supply decision-making and hindering their income increase. 

On one hand, although NRCMS has increased farmers’ willingness to transfer farmland to some degree, it actually reduces the amount of lands that transfer in and “lock-in” farmers in the farmland, which does not help the effective transfer and concentration, and affects the appropriate management of farmland [42]. On the other hand, as a social medical insurance scheme aiming at providing medical security for rural residents, NRCMS has limitations on participation and compensation. In particular, the complex procedures of reimbursement for medical treatment in non-registered places and the policy of graded compensation prevent these farmers from really enjoying the benefits brought by NRCMS, thus labeling the scheme with strong non-portability, which hinders the cross-regional migration of rural labor force and affects farmers’ income.

Importantly, the public health insurance may contribute to reducing poverty [6,54] and has a strong association on non-medical-related consumption [31] and private saving [3,5].

The government needs to consider more carefully whether the objectives of NRCMS are consistent with other relevant agricultural and rural development policies. First, the government should first simplify reimbursement procedures step by step, speed up the reimbursement platform construction in non-registered places, and promote the reform of payment systems like immediate reimbursement; second, on the basis of simplifying medical treatment in non-registered places and reimbursement settlement, the government should strive to improve the basic medical service capacity with endemic common diseases, frequently-occurring diseases, and graded diagnosis and treatment of chronic diseases, and guide the sinking of high-quality medical resources and form a reasonable medical system, which provides the basis for the establishment of graded diagnosis and treatment systems. Third, the government should gradually eliminate the regional division of the system itself, realize the sharing of medical resources and mutual recognition of diagnosis results, and improve the social-insurance-led, comprehensive, and portable medical insurance system. The improvement will make NRCMS more effective in adapting to the situation of cross-region and cross-urban migration of rural labor force, and realize the effective allocation and utilization of labor resources, which provides a significant prerequisite for orderly migration of rural labor force and increase of farmers’ income.

## 6. Conclusions

By adopting the data of CHARLS conducted in 2011 and 2013, this paper investigated the influence of NRCMS on farmers’ non-farm labor supply behavior. The main conclusions and enlightenment for policy are as follows:

Regarding the choice of non-farm labor participation, first of all, “whether to participate in NRCMS” will significantly reduce farmers’ non-farm labor supply willingness. Secondly, “whether to participate in supplementary medical insurance and reimbursement methods” is an important factor influencing farmers’ non-farm labor participation behavior. From the perspective of farmers’ choice of non-farm work types, NRCMS variables, such as participation or reimbursement methods, have significant influence with different directions, which to a large extent results from the graded system of NRCMS hospitalization compensation policy and the complicated reimbursement procedures for medical treatment in non-registered places.

After the further study of the non-farm labor time of farmers who participated in non-farm employment, it was found that the influence of NRCMS (whether to participate in NRCMS and whether to participate in supplementary insurance and reimbursement methods) is not significant, which needs to be further tracked and studied. Due to the limitation of data availability, this study only explored the influence of NRCMS on farmers’ non-farm labor supply behavior from a short-term perspective, but ignoring the long-term impact. In theory, the use of cross-temporal tracking data can more comprehensively reflect the influence of NRCMS on farmers’ non-farm labor supply behavior, which needs further studies. In addition, in recent years, China’s NRCMS has gradually implemented policies, such as in-patient compensation, general out-patient compensation, special out-patient compensation, and compensation for serious diseases. This study will analyze and discuss whether these policies influence farmers’ willingness to participate in non-farm labor and their actual behavior in order to better examine the relationship among NRCMS and the non-farm labor supply and the increase of farmers’ income.

## Figures and Tables

**Table 1 ijerph-16-04865-t001:** Descriptive statistics of variables.

Variable	Total	Employed (Participating Sample)	Non-Farm Self-Employed (Participating Sample)
Non-Farm Labor Supply			
Non-Farm Labor or Not	0.172 (0.377)	--	--
Annual Working Hours	--	1713.395 (1042.732)	1925.546 (1513.688)
NRCMS Participate in or Not (1 = Yes)	0.906 (0.291)	--	--
Supplementary Insurance (1 = Yes)	--	0.036 (0.186)	0.042 (0.200)
Reimbursement Methods (1 = Immediately; 0 = Self-payment before Reimbursement)	--	0.328 (0.470)	0.317 (0.466)
Individual Characteristics			
Age (Year)	60.638 (10.39)	54.283 (6.948)	55.755 (8.434)
Gender (1 = Male)	0.509 (0.499)	0.673 (0.469)	0.649 (0.477)
Marriage (1 = Married)	0.733 (0.443)	0.782 (0.413)	0.833 (0.373)
Primary School Education or above (1 = Yes)	0.486 (0.500)	0.672 (0.470)	0.681 (0.466)
Self-Evaluated Health (1 = Good)	0.699 (0.458)	0.849 (0.359)	0.827 (0.379)
Family Characteristics			
Family Size	3.298 (1.819)	3.220 (1.658)	3.341 (1.182)
Annual Net Income per Capita (10,000 yuan)	0.528 (0.654)	0.574 (0.505)	0.637 (2.015)
Owning Lands (1 = Yes)	0.798 (0.401)	0.817 (0.387)	0.799 (0.401)
Province Dummy Variable			
Fujian	0.039 (0.195)	0.064 (0.246)	0.076 (0.264)
Gansu	0.029 (0.169)	0.019 (0.138)	0.016 (0.125)
Guangond	0.036 (0.187)	0.061 (0.240)	0.042 (0.200)
Guangxi	0.038 (0.193)	0.028 (0.166)	0.044 (0.205)
Guizhou	0.014 (0.119)	0.007 (0.086)	0.006 (0.077)
Hebei	0.058 (0.233)	0.067 (0.251)	0.059 (0.237)
Henan	0.075 (0.264)	0.055 (0.229)	0.095 (0.294)
Heilongjiang	0.009 (0.095)	0.003 (0.055)	0.006 (0.077)
Hubei	0.039 (0.193)	0.045 (0.207)	0.034 (0.181)
Hunan	0.043 (0.203)	0.034 (0.182)	0.038 (0.191)
Jilin	0.014 (0.116)	0.007 (0.086)	0.010 (0.099)
Jiangsu	0.029 (0.169)	0.046 (0.210)	0.020 (0.140)
Jiangxi	0.037 (0.189)	0.052 (0.223)	0.048 (0.213)
Liaoning	0.035 (0.183)	0.042 (0.200)	0.036 (0.186)
Neimenggu	0.034 (0.180)	0.010 (0.101)	0.016 (0.125)
Qinghai	0.015 (0.120)	0.008 (0.086)	0.016 (0.125)
Shandong	0.088 (0.283)	0.138 (0.345)	0.062 (0.241)
Shangxi	0.040 (0.195)	0.032 (0.179)	0.042 (0.200)
Shanxi	0.034 (0.180)	0.043 (0.204)	0.040 (0.196)
Sichuan	0.100 (0.299)	0.052 (0.223)	0.068 (0.251)
Xinjiang	0.003 (0.059)	0.001 (0.038)	0.004 (0.063)
Yunnan	0.065 (0.248)	0.016 (0.127)	0.054 (0.226)
Zhejiang	0.046 (0.210)	0.103 (0.304)	0.094 (0.291)
Chongqing	0.020 (0.141)	0.013 (0.115)	0.008 (0.089)
Year 2013 (1 = Yes)	0.753 (0.431)	0.716 (0.451)	0.751 (0.433)
Observed Sample	8273	667	502

Note: Data in parentheses are standard deviation values. NRCMS: New Rural Cooperative Medical Scheme.

**Table 2 ijerph-16-04865-t002:** Estimates of the effect of New Rural Cooperative Medical Scheme (NRCMS) on non-farm labor supply.

Variable	Probability Model of Non-Farm Participation		Probability Model of Non-Farm Labor Type Choice	
	All Samples (1)	Participating Sample (2)	All Non-Farm Samples (3)	Participating Sample (4)
Participating in NRCMS	–0.016 *	--	–0.100 ***	--
Participating in Supplementary Insurance	--	–0.033 *	--	–0.010
Immediate Reimbursement	--	0.009 *	--	0.018
Age	–0.011 ***	–0.010 ***	–0.008 ***	–0.008 ***
Male	0.113 ***	0.118 ***	0.071 **	0.061 *
With Spouse	–0.011	–0.016 ***	–0.107 ***	–0.107 ***
Primary School Education or Above	0.041 ***	0.037 ***	–0.043	–0.056 *
Self-evaluated Health	0.061 ***	0.059 ***	–0.019	–0.017
Family Size	–0.005 **	–0.004 *	0.007	0.006
Annual Net Income per Capita	0.0002	0.0002	–0.019	–0.018
Owning Lands	–0.038 ***	–0.037 ***	0.076 ***	0.047
Year 2013	0.002	-0.004	–0.048 **	–0.048
Province Dummy Variable	Yes	Yes	Yes	Yes
Pseudo R2	0.1653 ***	0.1648 ***	0.0524 ***	0.0502 ***
Observed Sample	8273	7497	1441	1287

All models are marginal effects with robust standard errors in parentheses. * *p* < 0.10, ** *p* < 0.05, *** *p* < 0.001.

**Table 3 ijerph-16-04865-t003:** Estimates of the effect of NRCMS on non-farm labor time allocation.

Variable	Labor Time Model of the Employed: Ln (h)		Labor Time Model of the Non-Farm Self-employed: Ln (h)	
	All Employed Samples (1)	Participating Employed Sample (2)	All Non-Farm Self-Employed Samples (3)	Participating Self-Employed Sample (4)
Participating in NRCMS	–0.101	--	0.080	--
Participating in Supplementary Insurance	--	0.004	--	0.245
Immediate Reimbursement	--	0.050	--	0.018
Age	–0.011 **	–0.012 **	–0.029 ***	–0.029 ***
Male	0.105 *	0.135 *	0.100	0.083
With Spouse	0.003	0.0005	–0.019	–0.023
Primary School Education or Above	0.038	0.026	–0.106	–0.099
Self-evaluated Health	0.209 ***	0.219 **	0.044	0.059
Family Size	–0.045 **	–0.048 **	0.017	0.022
Annual Net Income per Capita	0.038	0.021	0.006	0.003
Owning Lands	–0.132 *	–0.161 *	–0.184 *	–0.184
Year 2013	0.078	0.097	–0.093	–0.108
Province Dummy Variable	Yes	Yes	Yes	Yes
R2	0.0445	0.0421	0.0523	0.0466
Observed Sample	758	667	549	502

All models are marginal effects with robust standard errors in parentheses. * *p* < 0.10, ** *p* < 0.05, *** *p* < 0.001.

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
