# Peer review of "Public Health Insurance, Non-Farm Labor Supply, and Farmers’ Income: Evidence from New Rural Cooperative Medical Scheme"

_ijerph, 2019, doi:10.3390/ijerph16234865_

Round 1
Reviewer 1 Report
REVIEWER’S COMMENTS ON ijerph-649127
In general, the paper is well written. I do enjoy the reading of this paper. I have the following comments and suggestions.
General Comments
(1) In the Introduction section, the writing of the research objectives and contributions is not very clear. I would suggest the authors could clearly point out them. The dataset and model used in this study should be briefly mentioned in the Introduction section.
(2) In Section 3.2, I would suggest the authors can re-arrange the order of the paragraphs. It is better that the authors could introduce the data source in the beginning before they present the information relate to data.
(3) Currently, the discussions on results, conclusions and policy implications are in a mess. I would suggest the authors could include a new Section such as “5. Conclusions and Policy Implications” to re-structure Section 4.2.
(4) When the authors present and discuss the empirical results, I would suggest the authors could compare their results with the findings in the literature.
(5) When preparing the manuscript, the authors should use the same font size. Currently, the font size is different, which should be corrected by the authors.
Some specific comments
Line 13 please use “Chinese farmers”;
Line 17 please replace “on” with “to”;
Line 19 please use “The empirical results show that NRCMS ….”
Line 21 please use “non-participants”;
Line 47 Citation is need after this sentence;
Line 125 please use “aims at examining the effect of …”
Line 320 please replace “Descriptive statistics” with “Descriptive results”;
Line 344 Please replace “influence” with “Impact”;
《THE END》
Author Response
Dear peer reviewer,
Thank you for these comments about our manuscript. Here are our responses to explain point-by-point the details to your comments.
(1) In the Introduction section, the writing of the research objectives and contributions is not very clear. I would suggest the authors could clearly point out them. The dataset and model used in this study should be briefly mentioned in the Introduction section.
Response 1: Thank you for the suggestion, we have clearly point out the research objectives and contribution on p.3 lines 114-123 in the revised manuscript. The dataset and model used in our research work also have be mentioned on p.3 lines 119-121.
(2) In Section 3.2, I would suggest the authors can re-arrange the order of the paragraphs. It is better that the authors could introduce the data source in the beginning before they present the information relate to data.
Response 2: We have re-arrange the paragraphs in Section 3.2 on p.7-8 lines 292-308.
(3) Currently, the discussions on results, conclusions and policy implications are in a mess. I would suggest the authors could include a new Section such as “5. Conclusions and Policy Implications” to re-structure Section 4.2.
Response 3: In this revised manuscript, we have re-structured Section 4.2, while including the new sections: Section 5. Conclusion and Section 6. Discussion and policy implication.
(4) When the authors present and discuss the empirical results, I would suggest the authors could compare their results with the findings in the literature.
Response 4: In Section 6, we discussed the empirical results on p. 13-14, lines 452-468 in the revised manuscript.
(5) When preparing the manuscript, the authors should use the same font size. Currently, the font size is different, which should be corrected by the authors.
Response 5: Thank you for pointing out the misplacement. We address and check the comment about the font size in revised manuscript.
(6) Some specific comments
Line 13 please use “Chinese farmers”;
Line 17 please replace “on” with “to”;
Line 19 please use “The empirical results show that NRCMS ….”
Line 21 please use “non-participants”;
Line 47 Citation is need after this sentence;
Line 125 please use “aims at examining the effect of …”
Line 320 please replace “Descriptive statistics” with “Descriptive results”;
Line 344 Please replace “influence” with “Impact”;
Response 6: Thank you for pointing out these misplacements. We address and check the comments about English language grammar and usage in revised manuscript. What’s more, we hired a professional English editor Sharon Lynn Bear for proofreading.
In this revised manuscript, Line 13 “Chinese farmers” has be used;
Line 17 It is done to replace “on” with “to”;
Line 19 “The empirical results show that NRCMS” is be used;
Line 21 The “non-participants” is be replace with “non-participants”;
Line 45 More literatures have been cited;
Line 126 We use “aims at examining the effect of…”;
Line 322 We use “Descriptive results”;
Besides, about the “influence” line 344 in the manuscript, we re-structure Section 4 “Empirical results” in this revised manuscript. And in Section 6, we cite some literature on labor supply in rural China on p.13 lines 452-454 in the revised manuscript.

Reviewer 2 Report
Manuscript Number: ijerph-649127
Title: Public Health Insurance, Non-farm Labor Supply and Farmer’s Income:
Evidence from New Rural Cooperative Medical Scheme
This paper mainly aims at investigating the impacts of the NRCMS on farmers’ non-farm labor supply behavior and farmers’ income. This topic is very interesting. The data and model are suitable. Therefore this research is of high actuality contributing to both ecological and practical purposes. Results of the investigation might be important and interesting for many researchers enchanting the understanding of impacts of NRCMS, and thus the manuscript might be recommended for publishing after minor revision. Some of my comments are as follows:
Comments:
The language should be improved. There are some awkward expressions or grammatical mistakes. E.g. line 23, “preliminary results” should be “results”. You should be confident with your results. Introduction sector: in the first paragraph, the font size is different. line37-38, “ the per capita disposable income of rural households has declined year by year”, this is right? In my opinion, the per capita disposable income of rural households increases gradually but not decreases. The share of agricultural income decreases gradually in recent years. Please confirm it. Please cite some literature to support the ideal. E.g., “Theatrically, as an important part of the social security system, the public health insurance 43 scheme helps to protect the physical and mental health of laborers, improve labor productivity, and 44 then promote development and increase residents’ income.” This sentence needs some citations. I suggest authors clearly address your contributions in Introduction sector. In my opinion, using “Theoretical Framework” as the title of the second part is more suitable. The equations should be edited. The section of Variables could be integrated into Methods section. Descriptive Statistics section could be integrated into Data section.
Author Response
Dear peer reviewer,
Thank you for these comments about our manuscript. Here are our responses to explain point-by-point the details to your comments.
The language should be improved. There are some awkward expressions or grammatical mistakes. E.g. line 23, “preliminary results” should be “results”. You should be confident with your results.
Response 1: Thank you for pointing out these misplacements. We address and check the comments about English language grammar and usage in revised manuscript. What’s more, we hired a professional English editor Sharon Lynn Bear for proofreading.
And on p.1 line 23, “preliminary results” has been replaced with “results”.
Introduction sector: in the first paragraph, the font size is different.
Response 2: We address and check the comment about the font size in revised manuscript.
line37-38, “ the per capita disposable income of rural households has declined year by year”, this is right? In my opinion, the per capita disposable income of rural households increases gradually but not decreases. The share of agricultural income decreases gradually in recent years. Please confirm it.
Response 3: Thank you for pointing out the misplacement. We correct it on p.1 lines 36-37 “on the other hand, the business income of per capita disposable income of rural households…”.
Please cite some literature to support the ideal. E.g., “Theatrically, as an important part of the social security system, the public health insurance 43 scheme helps to protect the physical and mental health of laborers, improve labor productivity, and 44 then promote development and increase residents’ income.” This sentence needs some citations.
Response 4: We cite some literature “then promote development and increase residents’ income [2-5]” on p. 1 line 43 in the revised manuscript.
I suggest authors clearly address your contributions in Introduction sector.
Response 5: Thank you for the suggestion, we have clearly point out the research objectives and contribution on p.3 lines 114-123 in the revised manuscript.
In my opinion, using “Theoretical Framework” as the title of the second part is more suitable. The equations should be edited.
Response 6: We use “Theoretical framework” on p.4 line 157 in the revised manuscript. The equations have been edited.
The section of Variables could be integrated into Methods section.
Response 7: In Section 3, the variables have been integrated on p.5-7 lines 247-290 in the revised manuscript.
Descriptive Statistics section could be integrated into Data section.
Response 8: The “Descriptive Statistics” section has been integrated into Data section on p.8 lines 322-343.

Reviewer 3 Report
abstract english need attention at one place. see line number 17. Tables need attention a table is not appriate in terms of its formatting. Also table 2 and 3 need alignement. As your data is longitudinal one is not clear whether you have used longitudinal data or just one wave of the data. Please make it clear Also, try to elaborate the data in more details. Furthermore, the font size is very small after first page of introduction which is surprising why is that please fix it?Author Response
Dear peer reviewer,
Thank you for these comments about our manuscript. Here are our responses to explain point-by-point the details to your comments.
1)abstract english need attention at one place. see line number 17.
Response 1: Line 17 It is done to replace “on” with “to” in this revised manuscript.
2)Tables need attention a table is not appriate in terms of its formatting. Also table 2 and 3 need alignement.
Response 2: The tables formatting including table 2 and 3 have be alignment.
3)As your data is longitudinal one is not clear whether you have used longitudinal data or just one wave of the data. Please make it clear Also, try to elaborate the data in more details.
Response 3: The China Health and Retirement Longitudinal Study (CHARLS) has received critical support from Peking University, the National Natural Science Foundation of China, the Behavioral and Social Research Division of the National Institute on Aging and the World Bank. CHARLS is a nationally representative longitudinal survey of persons in China 45years of age or older and their spouses, including assessments of social, economic, and health circumstances of community-residents [ Zhao, Y., Hu, Y., Smith, J. P., Strauss, J., and Yang, G. (2014), “Cohort profile: The China Health and Retirement Longitudinal Study (CHARLS)”, International Journal of Epidemiology, Vol. 43, No.1, pp. 61-68.]. All data will be made public one year after the end of data collection. CHARLS adopts multi-stage stratified PPS sampling. As an innovation of CHARLS, a software package (CHARLS-GIS) is being created to make village sampling frames.
The baseline national wave of CHARLS is being fielded in 2011 and includes about 10,000 households and 17,500 individuals in 150 counties/districts and 450 villages/resident committees (or villages) from 28 provinces. Furthermore, the CHARLS respondents are followed up every two years, using a face-to-face computer-assisted personal interview. Since 2011, the 2013 and 2015 and 2017 national follow-up survey data of CHARLS has been publicly released, and the 2017 survey data will be publicly. More detail could be logged on the website to download at http://charls.pku.edu.cn/pages/Data/111/en.
This point is being to expand at revised manuscript on p.7-8 lines 292-318.
4)Furthermore, the font size is very small after first page of introduction which is surprising why is that please fix it?
Response 4: We address and check the comment about the font size in revised manuscript.
Best wishes
